# *In situ* dynamic tracking of heterogeneous nanocatalytic processes by shell-isolated nanoparticle-enhanced Raman spectroscopy

Hua Zhang[1], Chen Wang[1], Han-Lei Sun[1], Gang Fu[1], Shu Chen[2], Yue-Jiao Zhang[1], Bing-Hui Chen[1], Jason R. Anema[1], Zhi-Lin Yang[2], Jian-Feng Li[1,2] & Zhong-Qun Tian[1]

Surface molecular information acquired *in situ* from a catalytic process can greatly promote the rational design of highly efficient catalysts by revealing structure-activity relationships and reaction mechanisms. Raman spectroscopy can provide this rich structural information, but normal Raman is not sensitive enough to detect trace active species adsorbed on the surface of catalysts. Here we develop a general method for *in situ* monitoring of heterogeneous catalytic processes through shell-isolated nanoparticle-enhanced Raman spectroscopy (SHINERS) satellite nanocomposites (Au-core silica-shell nanocatalyst-satellite structures), which are stable and have extremely high surface Raman sensitivity. By combining operando SHINERS with density functional theory calculations, we identify the working mechanisms for CO oxidation over PtFe and Pd nanocatalysts, which are typical low- and high-temperature catalysts, respectively. Active species, such as surface oxides, superoxide/peroxide species and Pd–C/Pt–C bonds are directly observed during the reactions. We demonstrate that *in situ* SHINERS can provide a deep understanding of the fundamental concepts of catalysis.

[1] MOE Key Laboratory of Spectrochemical Analysis and Instrumentation, State Key Laboratory of Physical Chemistry of Solid Surfaces, iChEM, College of Chemistry and Chemical Engineering, Xiamen University, Xiamen 361005, China. [2] Department of Physics, Research Institute for Biomimetics and Soft Matter, Xiamen University, Xiamen 361005, China. Correspondence and requests for materials should be addressed to J.-F.L. (email: Li@xmu.edu.cn) or to B.H.C. (email: chenbh@xmu.edu.cn) or to G.F. (email: gfu@xmu.edu.cn).

Understanding structure-activity relationships and reaction mechanisms is of significant importance in the rational design of highly efficient catalysts[1-5]. After some pioneering work on model systems by both surface and theoretical scientists[6-16], much progress has been made in this area. However, challenges associated with the expansion of model systems to industrial catalysts remain[8]. Thus, the development of *in situ* surface analysis techniques that can be used to study reaction processes occurring over practical catalysts is highly desirable. For example, some *in situ* techniques including environmental transmission electron microscopy, high-pressure scanning tunneling microscopy (STM), ambient pressure X-ray photoelectron spectroscopy (XPS) and X-ray absorption spectroscopy have been applied to study nanocatalysts and reveal molecular information about the catalysts under working conditions, such as surface compositions, chemical states, morphologies and coordination environments[17,18]. These works have greatly improved our understanding of catalysis.

One of the most widely used surface analysis techniques is *in situ* infrared (IR) spectroscopy, which can provide rich information about fine surface structure[19], active sites[20] and reaction mechanisms[21]. However, it is very difficult for *in situ* IR to detect intermediates with signals in the low wavenumber region of the spectrum. In contrast, the low wavenumber region is accessible by Raman spectroscopy[22]. This technique has been used to identify intermediate species such as metal-C bonds, active oxygen species, hydroxyl groups and surface oxides[23-25], but in general, normal Raman spectroscopy is not sensitive enough to monitor trace amounts of surface species on metal catalysts. Surface-enhanced Raman scattering (SERS) has been employed to circumvent this limitation, as it offers a tremendous improvement in sensitivity ($10^7$–$10^{10}$ times) and can even be used to detect single molecules[26-29]. Nevertheless, only a few metals

(primarily Au, Ag and Cu) with nanostructured surfaces provide a large SERS effect[30]. Thus, monitoring catalytic processes occurring over other metals remains a great challenge.

Recently, our group invented a novel technique known as 'shell-isolated nanoparticle-enhanced Raman spectroscopy' or 'SHINERS'[31], which has been regarded as a 'next generation of advanced spectroscopy'[32]. In SHINERS, plasmonic Au nanoparticles (Raman signal amplifiers) are coated with pinhole-free silica shells that prevent them from interacting with analytical targets or the chemical environment. The silica shells are thin enough (just a couple of nanometers) that the enhanced electromagnetic field generated at the surface of the core extends beyond the surface of the shell. SHINERS has broken the long-standing surface material and morphology restrictions that exist in traditional SERS. In principle, the SHINERS technique can be applied to a surface of any material and any morphology. It has been widely used in electrochemistry[31,33-35], life science[36], the materials and semiconductor industry[31,37], solar energy storage and conversion[38,39], and our daily life[40].

In this work, we develop a simple strategy to construct Au-core silica-shell nanocatalyst-satellite nanocomposites, which we call SHINERS-satellite structures. Three-dimensional finite-difference time-domain (3D-FDTD) simulations demonstrate that Raman signals from chemical species on the satellite structures are greatly amplified by the plasmonic activity of the core, making this an ideal architecture for monitoring heterogeneous catalytic reactions. With this strategy we have obtained direct spectroscopic evidence of the formation of metal-O and metal-C bonds, as well as active oxygen species, during CO oxidation. With the aid of density functional theory (DFT) calculations, the SHINERS-satellite method has revealed new molecular information about CO oxidation.

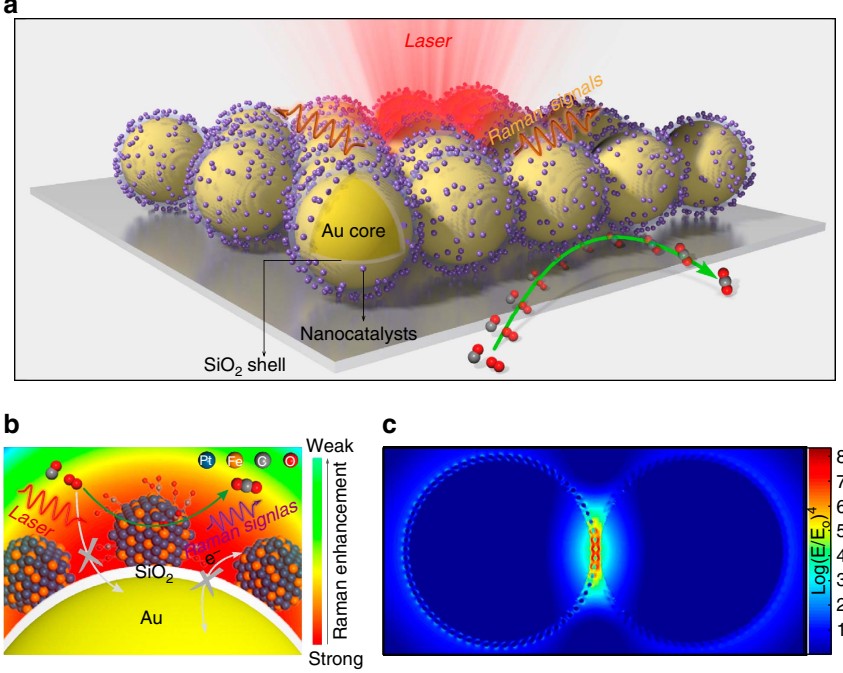

**Figure 1 | Schematic illustration of a SHINERS-satellite study of a nanocatalytic process.** (**a**) The Au-core silica-shell nanocatalyst-satellite architecture of SHIN-enhanced Raman spectroscopy (SHINERS)-satellite structures, and the mechanism for CO oxidation over PtFe bimetallic nanocatalysts revealed by our SHINERS-satellite method. The pinhole-free silica shell prevents the Raman signal amplifier from interfering with the system under study. (**b**) Schematic illustration of CO oxidation on PtFe. The blue, orange, grey and red spheres represent Pt, Fe, C and O atoms, respectively. (**c**) A 3D-FDTD simulation for a pair of Pt-on-shell isolated nanoparticle (SHIN) nanocomposite structures.

## Results

**Raman enhancement of SHINERS-satellite nanocomposites.** In SHINERS, light-induced localized surface plasmons yield areas of enhanced electromagnetic field strength at the surface of the Au core, and the electromagnetic field strength decays exponentially with distance from the surface of the Au core depending on near-field characteristics[31,41]. Thus, we designed a core–shell-satellite nanocomposite structure with the nanocatalysts on the silica shell, just a couple of nanometers away from the core. In this way, Raman signals from reaction intermediates on the surface of the catalysts can be amplified effectively.

The Au-core silica-shell nanocatalyst-satellite architecture is shown in Fig. 1a. The SHINERS-satellite structure is prepared by coating a Au nanoparticle with an ultrathin yet pinhole-free silica shell, then adding the nanocatalysts to the surface of the shell. The pinhole-free silica shell prevents adsorption of molecules from the chemical environment on the Au core, and it also prevents any interaction between the nanocatalysts and the Au core (Fig. 1b). Thus, information about heterogeneous catalytic reaction processes can be obtained without risk of interference by the Raman signal amplifier. To confirm that this structure will effectively enhance Raman signals from species on the nanocatalysts, we modelled electromagnetic field strength using a 3D-FDTD method in which the perfectly matched layer boundary conditions were adapted to avoid non-physical reflections[42]. The structure selected for modelling was a 120 nm Au core with a 2 nm silica shell supporting 2 nm Pt nanocatalysts. The dielectric functions of Au and Pt were taken from a multi-coefficient fitting model offered by Lumerical Solutions. Figure 1c shows that a region of extraordinarily high electromagnetic field strength (a so-called 'hotspot') occurs in the nanoparticle-nanoparticle junction when a pair of SHINERS-satellite structures are illuminated with a 633 nm laser (the total-field scattered-field source acted as a linearly polarized light normal to the pair). The 3D-FDTD simulations show that Raman scattering from molecules on the nanocatalysts located in the junction between the two shell-isolated nanoparticles (SHINs) would be enhanced by up to 8 orders of magnitude, which is strong enough for ultrasensitive detection of surface species[33].

**Synthesis of SHINERS-satellite nanocomposites.** The nanocatalyst satellites are added to the surface of the SHINs by a self-assembly process. The synthesis of our SHINs is described in detail elsewhere[43]. After synthesis, the SHIN surface is negatively charged due to the presence of citrate (Supplementary Fig. 1). The PtFe bimetallic nanocatalyst, which is widely used for oxygen reduction reactions in fuel cells[44] and catalytic oxidation[45], was selected as a representative catalyst with which to explore the synthesis of SHINERS-satellite nanostructures. The synthesis and characterization of the nanocatalysts is detailed in the Supplementary Methods. The PtFe nanocatalysts were modified to create a positively charged surface (Supplementary Figs 1–3), and mixed with the negatively charged SHINs (Supplementary Fig. 1) to achieve the self-assembly process. Figure 2a clearly shows that the resulting nanoparticles have a core–shell-satellite tiered architecture. This structure is further demonstrated by the element maps in Fig. 2b and the high-resolution transmission electron microscopy (HR-TEM) images in Supplementary Fig. 4. In a control experiment, unmodified PtFe nanocatalysts were added to a solution of SHINs. They did not adhere to the silica surface, demonstrating that modification of the PtFe nanocatalysts to create a positively charged surface is critical for SHINERS-satellite synthesis (Supplementary Fig. 5).

Through a systematic study, we found that the strategy employed here for PtFe nanocatalysts can be used to assemble many other types of nanocatalyst-on-SHIN structures (Supplementary Fig. 6). Figure 2c shows HR-TEM images of monometallic Pt and Pd nanocatalysts, bimetallic PtPd and PtFe nanocatalysts, Au@PtFe core–shell structures, PdFeCu nanocubes, and $CeO_2$ and $Fe_2O_3$ metal oxides on SHINs. The size and morphology of these nanocatalysts were unchanged by the self-assembly process, and all of them were evenly distributed on the surface of the SHINs. Thus, it is feasible to study structure-activity relationships and reaction mechanisms of heterogeneous catalytic processes by SHINERS after carefully manipulating the composition, structure, morphology and size of the nanocatalyst satellites.

**CO oxidation over PtFe bimetallic nanocatalysts.** SHINERS-satellite structures were then used in operando studies of CO oxidation over Pt-group nanocatalysts. This process is of great importance in fundamental research, and has been recognized as a benchmark system for heterogeneous catalysis[46]. It is also important for practical applications such as environmental protection[47] and the production of ultrapure hydrogen[48]. The SHINERS-satellite method was first used to examine the relationship between structure and activity for PtFe nanocatalysts. Recent studies have shown that PtFe bimetallic nanocatalysts are highly active in the oxidation of CO (refs 45,49). DFT calculations indicate the high activity results from efficient activation of gaseous $O_2$ molecules by the coordinatively unsaturated Fe centres in PtFe (ref. 45). Direct spectroscopic evidence is still needed to prove this mechanism. The *in situ* SHINERS-satellite strategy introduced above can reveal the effect of the ferrous centres on the performance of CO oxidation.

The valence state of Fe in the PtFe nanocatalyst was determined by XPS (green data points in the lower part of Fig. 3a). For reference, XPS spectra of Fe in different oxidation states were obtained by controlling the sampling depth via $Ar^+$ sputtering (blue, red and black curves in the upper part of Fig. 3a). Fe foil exposed to the atmosphere was oxidized while the bulk material remained in metallic form. The green data points in Fig. 3a show an Fe $2p_{3/2}$ peak at ∼710.4 eV. We performed a deconvolution of the XPS spectrum to determine the chemical states of the Fe species (lower part of Fig. 3a) according to the literature[50] and the reference data obtained for Fe foil with different $Ar^+$ sputtering times. The fit results show that $Fe^{2+}$ species, as well as a very small amount of $Fe^0$ species, are present in the bimetallic catalysts. It can also be observed that the binding energies for these species are slightly higher than those in bulk Fe foil. This may be due to the lower Fe–Fe coordination and higher Fe–Pt coordination in the bimetallic catalyst, which would lead to pronounced electronic interactions between Fe and Pt (refs 51,52). An up-shift of the X-ray diffraction pattern of PtFe compared to Pt indicates a shrinking of the crystal lattice, meaning that Fe forms an alloy with Pt (Supplementary Fig. 7). Furthermore, element maps and line scans show that Fe species locate very close to Pt species (Supplementary Fig. 8). Therefore, we believe the PtFe bimetallic catalyst has an alloy structure with ferrous oxide on its surface.

Figure 3b compares the performance of the PtFe bimetallic nanocatalyst and a Pt monometallic nanocatalyst for CO oxidation at different temperatures. Clearly, the PtFe catalyst is much more active than the Pt catalyst. CO is completely removed by PtFe, even at room temperature. The PtFe nanocatalyst can also be used for preferential oxidation of CO in the presence of $H_2$, which is of significant importance for the production of highly pure $H_2$. As shown in Supplementary Fig. 9, the PtFe catalyst is highly active and stable in the preferential oxidation of CO at room temperature.

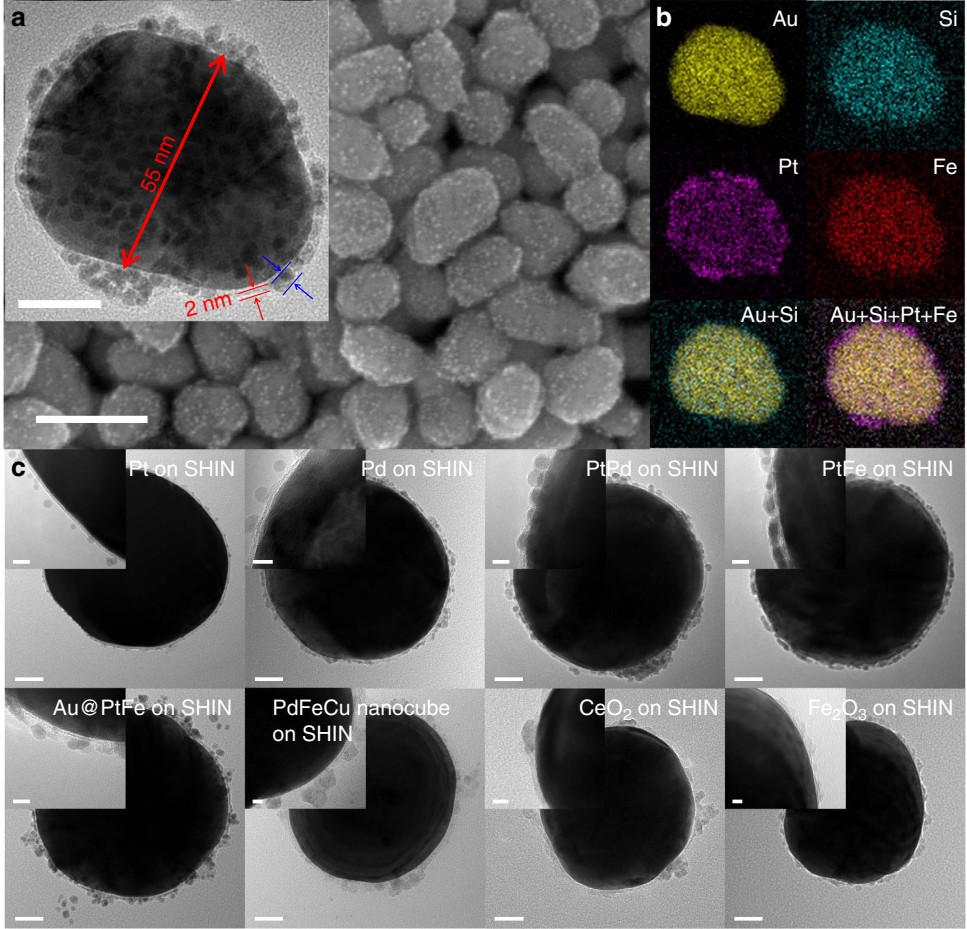

**Figure 2 | The structure of various SHINERS-satellite nanocomposites.** (**a**) TEM (inset) and scanning electron microscope images of PtFe-on-shell isolated nanoparticle (SHIN) core–shell-satellite nanocomposites. Scale bar, 100 nm. Scale bar for the inset, 20 nm. (**b**) Element maps of the single particle in the inset of (**a**). (**c**) TEM images of various nanocatalyst-on-SHIN structures. The insets in **c** are the zoomed-in image of the edge of the same particle as in main image. Scale bars, 20 nm. Scale bars for the insets, 5 nm.

To reveal the effect of the ferrous centres, PtFe-on-SHIN and Pt-on-SHIN nanocomposites were prepared for *in situ* SHINERS-satellite studies. Activity tests (Supplementary Fig. 9) and XPS data (Supplementary Fig. 10) show that the silica shell isolates the Au core so that it has no influence on the properties of the nanocatalysts. Furthermore, the capping agents on the catalysts (such as oleylamine) were removed during fabrication of the SHINERS-satellite nanocomposites and the preparation steps carried out before *in situ* SHINERS-satellite studies (Supplementary Fig. 11), so that catalytic sites were accessible to reactants. The black and pink curves in Fig. 3c show that no Raman signals are observed for Pt or PtFe nanocatalysts on silica due to the low sensitivity of normal Raman spectroscopy. The green curve shows that there are also no Raman signals for SHINs without nanocatalyst satellites, and therefore, the signals obtained in the following experiments must be from species adsorbed on the catalyst surface. The blue and red curves in Fig. 3c show that strong Raman peaks can be obtained from the same catalysts when they are incorporated into the SHINERS-satellite structures. These results demonstrate the role that SHINs play in Raman signal amplification. We note that the SHIN-satellite nanostructures are also much more stable than Au-satellite nanostructures at high temperatures (Supplementary Fig. 12). The two peaks at 397 and 485 cm$^{-1}$ in the spectrum obtained from the Pt-on-SHIN structures (blue curve) can be assigned to the Pt–C stretching vibrational mode of bridge and linear adsorbed CO respectively[53]. This spectrum (the blue curve) indicates that only

CO was adsorbed on the surface of the Pt, and the activation of $O_2$ was inhibited by the strongly competitive adsorption of CO. This can also be demonstrated by *in situ* SHINERS-satellite studies of Pt under pure $O_2$ and under CO oxidation conditions at higher temperatures (Supplementary Fig. 13).

The spectrum obtained from the PtFe-on-SHIN structures (Fig. 3c red curve) has peaks for $O_2^{2-}$ at 870 and 951 cm$^{-1}$, and a peak for $O_2^-$ at 1,158 cm$^{-1}$ (refs 23–25), in addition to the CO adsorption peaks at 389 and 480 cm$^{-1}$. The presence of active oxygen species on the PtFe nanoalloy surface was confirmed by electron paramagnetic resonance (EPR) studies after CO oxidation (Supplementary Fig. 14). It can be seen that there is no peak in the 500–700 cm$^{-1}$ range, suggesting that no Pt oxide is formed at 30 °C. Furthermore, a red shift of the Pt–C stretching vibration occurs for the PtFe nanoalloy compared to the Pt monometallic nanocatalyst (Fig. 3c inset). This implies that Fe species weaken the adsorption of CO, and this may be another reason for the high activity of the PtFe nanoalloy. These results indicate that CO oxidation on the PtFe nanoalloy could proceed through a Langmuir–Hinshelwood mechanism, even at very low temperature, in which the adsorbed CO and surface oxygen species are involved.

**CO oxidation over Pd nanocatalysts**. Pd based nanoparticles in catalytic converters are effective CO oxidation catalysts and reduce automotive emissions, however, the underlying

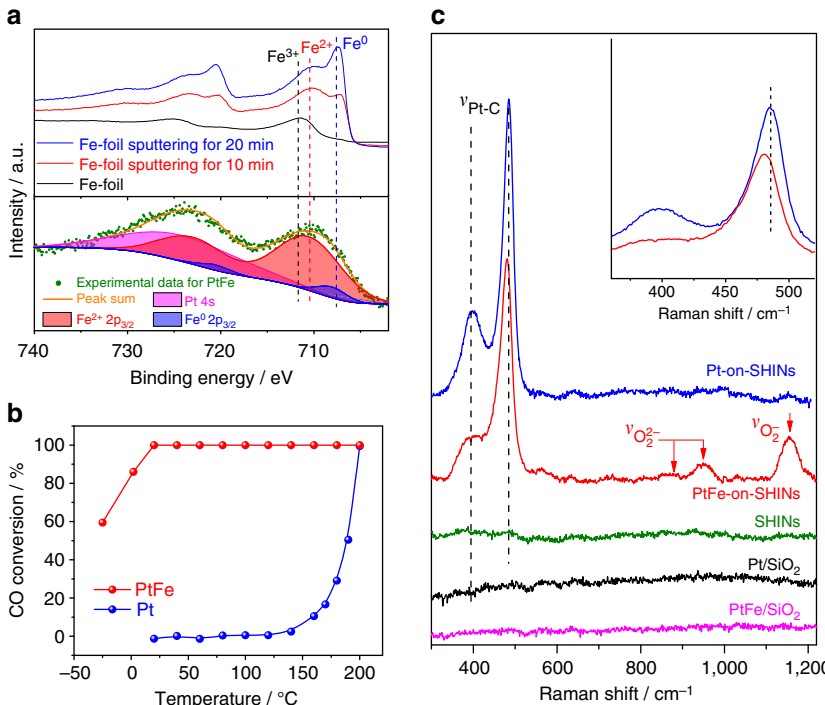

**Figure 3 | SHINERS-satellite study of CO oxidation over PtFe bimetallic nanocatalysts.** (**a**) XPS study to determine the valence state of Fe in the PtFe nanoalloy. (**b**) Catalytic performance of the PtFe nanoalloy and a Pt monometallic nanocatalyst for CO oxidation at different temperatures. (**c**) SHINERS-satellite spectra of CO oxidation over the PtFe nanoalloy and the Pt monometallic nanocatalyst at 30 °C. The inset of **c** shows that a red shift of the Pt–C stretching vibration occurs for the PtFe nanoalloy compared to the Pt monometallic nanocatalyst.

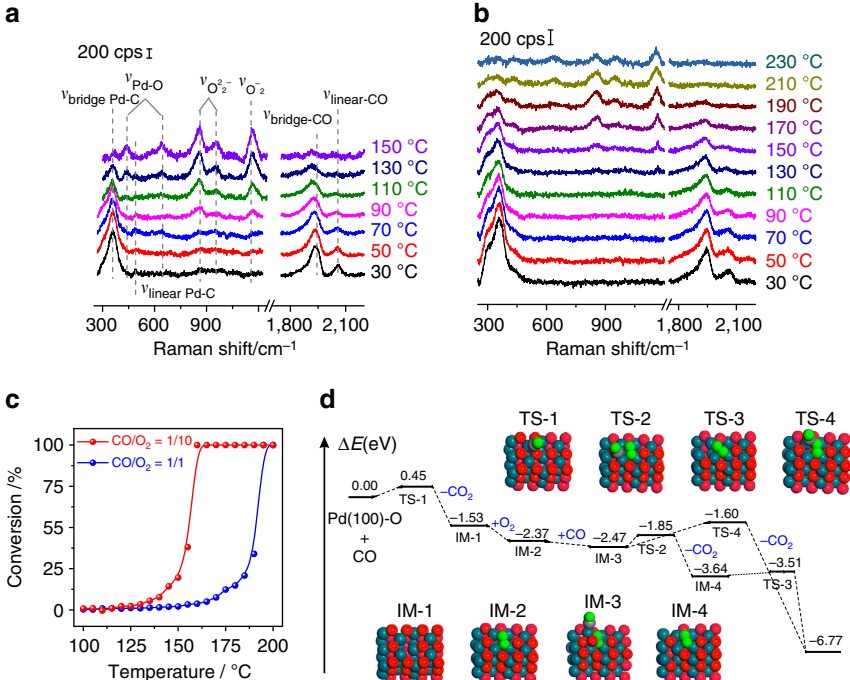

**Figure 4 | A SHINERS-satellite study and DFT calculations for CO oxidation over Pd nanocatalysts.** (**a,b**) SHINERS-satellite spectra for CO oxidation over Pd nanocatalysts with different gas ratios in the feed: (**a**) CO/O$_2$=1/10, (**b**) CO/O$_2$=1/1. (**c**) Catalytic performance for CO oxidation over Pd nanocatalysts under different feed conditions. (**d**) Mechanisms for CO oxidation over a PdO(100)–O surface.

mechanism is not fully understood. Figure 4a provides SHINERS-satellite spectra for CO oxidation over Pd nanocatalysts between 30 and 150 °C. At low temperatures ($\sim$30–50 °C), there is one strong peak in the low wavenumber region at 360 cm$^{-1}$ as well as

a very small peak at 490 cm$^{-1}$, and two peaks in the high wavenumber region at 1,935 and 2,061 cm$^{-1}$. The peaks at 360 and 1,935 cm$^{-1}$ can be assigned to bridge adsorbed CO on Pd, and the peaks at 490 and 2,061 cm$^{-1}$ can be assigned to linear

adsorbed CO on Pd (ref. 54). At higher reaction temperatures the intensity of these bands decreases, indicating that CO starts to desorb from the surface. When the temperature is increased above 70 °C, other peaks appear in the low wavenumber region. As shown in Supplementary Fig. 15, the species responsible for the additional bands at 860, 955 and 1,165 $cm^{-1}$ can be eliminated by replacing $O_2$ in the feed with $H_2$, and are restored by the reintroduction of $O_2$. Therefore, these bands must be related to surface oxygen species. The existence of surface oxygen species on Pd nanocatalysts was confirmed by XPS (Supplementary Fig. 16) and EPR (Supplementary Fig. 17). According to the literature[23–25], the bands at 860 and 955 $cm^{-1}$ can be assigned to peroxide species while the band at 1,165 $cm^{-1}$ can be assigned to superoxide species. These bands can also be assigned via DFT calculations (Supplementary Table 1), which will be discussed in detail later. When the temperature is further increased above 110 °C, two peaks appear at 433 and 645 $cm^{-1}$. These bands can be attributed to Pd–O stretching vibrations[55]. Their Raman shifts are slightly higher than the Raman shifts obtained from bulk Pd–O (Supplementary Fig. 18), and HR-TEM shows that no bulk PdO is formed during our Pd-on-SHIN experiments at 150 °C (Supplementary Fig. 19). These results indicate that the 433 and 645 $cm^{-1}$ peaks are due to surface oxides and not bulk PdO.

We also studied the influence that the $CO/O_2$ gas ratio has on surface species during CO oxidation (Fig. 4 and Supplementary Fig. 20). The SHINERS-satellite spectra obtained with different $CO/O_2$ gas ratios in the feed are very similar, indicating that CO oxidation processes on Pd nanocatalysts under different feed conditions are similar. At low temperatures, only CO is adsorbed on the catalyst. As reaction temperature increases, the Raman signal intensity increases for oxygen species and $PdO_x$ while the Raman signal intensity decreases for adsorbed CO species. These results indicate that at high temperatures, oxygen species and $PdO_x$ replace some of the CO on Pd. When the fraction of CO in the feed increases, the temperature when surface oxygen species and $PdO_x$ appear as well as the ratio of the Raman intensity of adsorbed CO to their intensity increases. At the same time, the activity of CO oxidation decreases with the increase of the $CO/O_2$ ratio. This indicates that CO inhibits activation of $O_2$ on the catalyst surface, and it can be explained by the fact that the adsorption energies for CO on Pd are more negative than those for $O_2$ on Pd (Supplementary Table 1 and 2). Therefore, the catalyst surface is preferentially covered by CO when exposed to a mixture of CO and $O_2$.

SHINERS-satellite studies can provide further information about the activity results obtained for CO oxidation on Pd (Fig. 4 and Supplementary Fig. 20). Figure 4c and Supplementary Fig. 20d plot catalytic performance against temperature for CO oxidation over Pd with different $CO/O_2$ gas ratios. At low temperatures, only CO is adsorbed on the catalyst and almost no activity is observed. When the temperature increases, peroxide and superoxide bands appear and the activity increases, indicating that these species are essential for CO oxidation. Pd–O bands are clearly observed under conditions of high activity, providing further evidence that $PdO_x$ is active. The adsorption of CO is not a prerequisite for CO oxidation because the Pt–C and CO bands nearly disappear for $CO/O_2$ ratios of both 1/10 and 1/1 at high temperatures where catalytic performance is high (Fig. 4a,b). This indicates that CO oxidation on Pd under oxygen rich conditions could occur via the Eley-Rideal mechanism, which is dramatically different from what occurs on PtFe (Supplementary Fig. 21). It should be noted that our findings lend support to the mechanism proposed by Frenken and co-workers[56], who claimed that the active surface is Pd oxide under $O_2$-rich conditions. They found that $O_2$ could be adsorbed on the surface to generate a hyperactive state, and that the $O_2$

could react with gaseous CO to yield $CO_2$. It can also be observed that the catalytic efficiency is lower when there is a greater fraction of CO in the feed (Fig. 4c and Supplementary Fig. 20d). This is because CO blocks $O_2$ adsorption, and higher temperatures are required to form active oxygen species and surface $PdO_x$ at higher $CO/O_2$ gas ratios. It is especially interesting that both CO and $O_2$ are present on the catalyst surface under the hyperactive state for $CO/O_2 = 5/1$ (Supplementary Fig. 20c). This means CO oxidation over Pd will change from a primarily Eley-Rideal mechanism to a primarily Langmuir–Hinshelwood mechanism when the feed changes from $O_2$-rich conditions to CO-rich conditions.

DFT calculations were carried out to better understand the mechanism of CO oxidation over Pd based catalysts. The different oxidation states of Pd were modelled as Pd(111), Pd(111)-O, $Pd_5O_4$ and PdO(100)–O (Supplementary Fig. 22). Adsorption energies, and other quantities associated with adsorption, are provided in Supplementary Tables 1 and 2. On Pd(111) and Pd(111)-O, CO adsorption is favoured over $O_2$ adsorption, which accounts for the fact that CO adsorption is dominant at low temperatures (<70 °C). As the temperature increases, surface Pd oxides begin to form. DFT calculations show that the adsorption of $O_2$ and CO become competitive on $Pd_5O_4$ (−0.64 eV and −0.76 eV respectively), but neither $O_2$ nor CO can be adsorbed on PdO(100)–O. We also consider the adsorption of $O_2$ on oxygen vacancies because oxygen vacancies may enhance CO oxidation on Pd oxide surfaces[57]. Here, two stable $O_2$ adsorption configurations, namely $O_2$(o) and $O_2$(c) (o and c refer to open and closed vacancies), are possible. Low spin density (0.00–0.38) and low O–O bond frequencies (806–944 $cm^{-1}$) indicate that $O_2$(o) and $O_2$(c) on the vacancies of both $Pd_5O_4$ and PdO(100)–O can be categorized as peroxide species. Compared to CO adsorption, $O_2$ adsorption on the oxygen vacancies is competitive, especially under excess oxygen conditions.

Mechanisms for the $2CO + O_2 \rightarrow 2CO_2$ process over PdO(100)–O are given in Fig. 4d. Since the vacancy free Pd(100)–O surface is repulsive towards CO, direct interaction between gas phase CO and a lattice O atom via the Eley-Rideal mechanism is the only possibility[58]. Figure 4d shows that the transition from CO and PdO(100)–O over TS-1 to $CO_2$ and IM-1, which has an oxygen vacancy, requires a small activation energy of 0.45 eV and is strongly exothermic. The resulting oxygen vacancy can be occupied by $O_2$ (IM-2), which makes the process exothermic by an additional 0.84 eV. The CO can weakly adsorb on the nearby Pd (IM-3), and then two different routes are possible. On one hand, the second CO can attack a lattice oxygen atom to form $CO_2$ and another oxygen vacancy, then the peroxide species would undergo O–O bond cleavage and recover the oxygen vacancy (Supplementary Fig. 23). Our calculations show that the transition from IM-3 over TS-2 to IM-4 presents a small barrier of 0.62 eV, and the transition from IM-4 over TS-3 presents a barrier of only 0.13 eV. On the other hand, the second CO can react with a peroxide species, leading to the formation of $CO_2$ and an oxygen vacancy free surface. However, a barrier of 0.87 eV must be surmounted to pass from IM-3 over TS-4. Based these results, we propose that the real active oxygen species on the PdO(100)–O surface are the lattice oxygen atoms, while the peroxide and superoxide species would readily dissociate to regenerate the lattice oxygen atoms, thus closing the catalytic cycle.

## Discussion

The *in situ* SHINERS-satellite method demonstrated in this work can be used to identify surface species with vibrational modes in

the low wavenumber region. When combined with more traditional characterization techniques such as TEM, XPS, X-ray absorption spectroscopy and IR, as well as DFT calculations, the SHINERS-satellite strategy can provide a deep understanding of structure-activity relationships and reaction mechanisms.

This strategy can be used to track surface species and intermediates of other industrially important reactions, besides the model CO oxidation reaction. For example, adsorbed ethylene species are easily detected by the SHINERS-satellite method (Supplementary Fig. 24), so that epoxidation of ethylene can be studied *in situ*. Another important aspect of the SHINERS-satellite strategy is its suitability for liquid phase reactions. Only Raman signals from species located a few nanometres from the Au core are enhanced, thus Raman signals from solvent molecules and other species in the bulk solution which do not participate in the reaction are not enhanced. The SHINERS-satellite method can also be used to characterize the composition and electronic properties of a catalyst surface if probe molecules are present (Supplementary Fig. 25). Furthermore, the Raman signal enhancement can be improved as necessary by optimizing the amplifier (for example, by increasing the Au core size or changing the Au core to a Ag one as shown in Supplementary Fig. 26)[40,59] to detect species with smaller Raman scattering cross-sections. These unique properties of the SHINERS-satellite strategy make it a universal and sensitive approach for *in situ* study of various reactions occurring on different nanocatalysts.

To summarize, a simple, general and stable nanocomposite strategy has been developed to monitor heterogeneous catalytic processes *in situ* with extremely high surface sensitivity using Au-core silica-shell nanocatalyst-satellite structures. 3D-FDTD simulations show that Raman signals from species on the surface of the nanocatalysts can be amplified by 8 orders of magnitude because of electromagnetic field enhancement by the Au cores. The silica shells isolate the Au cores and prevent them from interacting with the nanocatalysts and the chemical environment while improving their thermal stability. Using this strategy, we studied the oxidation of CO on PtFe nanoalloys and Pd nanocatalysts. For the PtFe system, we obtained direct spectroscopic evidence showing that the ferrous centre can weaken the Pt–C bond and activate $O_2$ at room temperature. This leads to CO oxidation by the Langmuir–Hinshelwood mechanism. For the Pd system, we combined the SHINERS-satellite strategy with DFT calculations and found that active $O_2$ species are not formed until CO begins to desorb, causing the reaction to follow the Eley-Rideal mechanism. We have demonstrated that SHINs may be combined with a variety of other nanocatalysts as well, meaning that our strategy can be developed as a standard characterization method for *in situ* monitoring of reaction intermediates and elucidating reaction mechanisms.

## Methods

**Synthesis of SHINs and nanocatalysts.** SHINs with 55 nm Au cores were synthesized according to our previously reported method[31]. In brief, 55 nm Au nanoparticles were synthesized according to Frens' method[60] and the Au nanoparticles were then coated with an ultrathin silica shell at 90 °C using 3-aminopropyltrimethoxysilane as a coupling agent and sodium silicate as a silicon source. SHINs with 120 nm Au cores were synthesized by a similar procedure, but 120 nm Au nanoparticles were prepared by a seed-mediated growth method and details of that synthesis can be found in the Supplementary Information. Procedures for the preparation of nanocatalysts such as Pt, Pd, PtPd, PtFe, Au@PtFe, PdFeCu nanocubes, $CeO_2$ and $Fe_2O_3$ are also described in the Supplementary Methods. Pt, Pd and PtFe nanoparticles were supported on $SiO_2$ for catalytic tests, and the loading of metal was controlled to ∼2.5 wt% as measured by inductively coupled plasma-atomic emission spectrometry (ICP-AES).

**Self-assembly of nanocatalyst-on-SHIN structures.** In general, 1 ml of 0.1 M $NOBF_4$ in acetonitrile was added to each 0.5 ml dispersion of as-prepared

nanocatalysts in toluene. Each mixture was shaken vigorously for 5–10 min. Hexane and more toluene were added to precipitate the nanocatalysts, and the separation was completed by centrifuging. The supernatant was discarded, and acetonitrile was added to form a clear dispersion of the nanocatalysts. By this procedure, the nanocatalysts were transferred from toluene to acetonitrile and their surfaces were modified to be positively charged as indicated by zeta potential measurements (Supplementary Figs 1 and 6). SHINs dispersed in water were then added, and each mixture was shaken overnight to complete the self-assembly process. Each mixture was centrifuged one last time, and the supernatant was discarded to remove excess nanocatalyst. The resulting precipitate was the desired Au-core silica-shell nanocatalyst-satellite nanocomposite, and was dispersed in acetonitrile for further use.

**Characterization.** Imaging and elemental analysis of these nanomaterials were accomplished by HR-TEM coupled with energy dispersive X-ray spectrometry using an FEI Tecnai F30 microscope. The bulk composition of the samples was determined by inductively coupled plasma-atomic emission spectrometry using an IRIS Intrepid II XSP spectrometer. XPS measurements were made using a VG MultiLab 2000 spectrometer with an Omicron Sphera II hemispherical electron energy analyser. X-ray diffraction patterns were recorded on a Rigaku Ultima IV with Cu Kα radiation operating at 40 kV and 30 mA. EPR spectra were acquired with a Bruker EMX 10/12 X-band spectrometer.

**In situ SHINERS-satellite studies.** About 5 μl of Au-core silica-shell nanocatalyst-satellite structures in acetonitrile were deposited on a Si substrate and dried at room temperature. The substrate was then placed in a Raman cell with reaction temperature and gas flow control (the cell was made in-house). The samples were held at 30 °C under $H_2$ for 30 min to remove pre-adsorbed oxygen species and other contaminants. For *in situ* SHINERS-satellite studies of CO oxidation, the reaction temperature varied from 30 to 150 °C and the reaction gas consisted of 1% CO, 10% $O_2$ and 89% $N_2$. The SHINERS-satellite experiments were carried out using a Jobin-Yvon Horiba Xplora confocal Raman system. The × 50 microscope objective had a numerical aperture of 0.55 and a power density of ∼1.5 mW um$^{-2}$. The performance of the catalysts was measured in a fixed-bed lab reactor at atmospheric pressure with a weight hourly space velocity of 40,000 ml g$^{-1} \cdot$ h$^{-1}$.

**Data availability.** The authors declare that all the data supporting the findings of this study are available within the paper and its Supplementary Information or from the corresponding author upon reasonable request.

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

## Acknowledgements

This work was supported by the NSFC (21522508, 21427813, 21373167, 21521004, 21573178 and 21673187), Natural Science Foundation of Guangdong Province (2016A030308012), the Fundamental Research Funds for the Central Universities (20720150039 and 20720160046), '111'Project (B16029), and the Thousand Youth Talents Plan of China. We thank Dr C. Liu and M. Meng for experimental support.

## Author contributions

H.Z. and J.-F.L. designed the experiments. H.Z., H.-L.S., C.W. and Y.-J.Z. carried out the experiments. G.F. conducted the DFT calculations, and S.C. and Z.-L.Y. conducted the 3D-FDTD simulations. H.Z., J.R.A., B.-H.C., J.-F.L. and Z.-Q.T. analysed the data. All authors contributed to the preparation of the manuscript.

## Additional information

**Competing interests:** The authors declare no competing financial interests.

