## [Peer Review File · Nature Communications]

Reviewers' comments:

Reviewer #1 (Remarks to the Author):

This manuscript describes measurement of surface enhanced Raman spectra (SERS) during CO oxidation catalysis on metal catalysts (E.g, Pt, FePt, Pd) supported on ~55nm Au nanoparticles coated with a 2nm thick SiO₂ layer. The Au nanoparticles create surface plasmon resonance (SPR), which enhances the Raman signal, while the SiO₂ layer acts as an inert catalyst support and isolates the Au from reactants and catalysts. This SiO₂ shell isolation technique allows extending the SERS to general systems beyond metals like Au that exhibit SPR, which has been described in detail in previous publications from the authors. The novelty of the present work is to demonstrate that the vibrations from metal-C bonds and various surface oxygen species (oxo, peroxy, superoxy) during CO oxidation can be probed using this technique, which allows to confirm certain differences in reaction mechanisms on different catalysts. The prepared materials are characterized using microscopy and various spectroscopic methods, and some conclusions are supported by DFT calculations. This work is of general interest in catalysis for the detection of low frequency vibrational modes during reactions, and is publishable if the issues mentioned in the comments below can be addressed.

The presence of surface oxygen species detected by the authors is qualitatively consistent with the proposed mechanism. However, these measurements were performed at one fixed reactant concentration in the feed, and only one temperature for Pt and FePt. A broader range of reaction conditions must be used to demonstrate, which of the species concentrations change linearly with reaction rates, or in a manner expected from proposed mechanism.

It seems likely to this reviewer, that the material synthesis methods leave some strongly bound chemical species on the catalyst surfaces. It is not clear if some treatment was used to remove those species and a complete access of catalyst sites to reactants was confirmed using chemisorption or any other methods.

In the last paragraph of page 8 the reviewers mention that 360 cm⁻¹ peak on Pd is from Pt-C vibrations of bridge CO, but a corresponding peak from linear CO with the same intensity ratio as that for the C-O stretch for the two species seems missing from the spectra. Is there a reason why this peak is missing?

The portion of spectra and conditions shown for the Pt and FePt samples are different from that shown for the Pd samples. The Supporting Information on comparing them at same spectral features and details of reaction dynamics should be provided to confirm internal consistency.

The last paragraph on page 7 mentions that the "Raman enhancements can be preserved even after high-temperature thermal treatments (Fig. S9)." The temperature of the treatment should be mentioned in the main text, and a comparison of peaks at same conditions before and after thermal treatment should be provided in the SI.

The abbreviation SHINERS is not defined the first time it is used in the abstract.

Reviewer #2 (Remarks to the Author):

Review of Nature Communications, NCOMMS-16-25667

"In situ tracking of heterogeneous nanocatalytic processes..."

Overall, this is a very nice paper that describes the application of a relatively new and novel method for *in situ* characterization of catalytic particles. These types of studies are very important as one piece of the puzzle for understanding heterogeneous catalysis. Although the authors claim that they are using this technique to “understand” the catalysis so as to facilitate rational design, the technique by itself is limited because it only probes vibrations of species on the surface; thus, it does not directly “see” changes in the catalyst structure or composition. I strongly suggest that the authors note that their new tool is a step forward but that it needs to be combined with one or more *in situ* method (e.g. ambient pressure XPS, X-ray absorption or environmental TEM) to get a full picture of the catalyst. In other words, it is important to point out that one technique cannot solve the problem of rational design. I think this paper would be a great choice for *Nature Communications* once revised to address the general point above and the technical details below.

1. The authors focus entirely on CO oxidation, one of the simplest catalytic reactions. Some discussion of how generalizable this method beyond CO to other systems that may not have such strong Raman signals should be added. For example, would it be possible to study ethylene or propylene epoxidation, methanol selective oxidation, or other important reactions? What are the limits of temperature and pressure for this technique? Adding such a discussion would increase the impact of the paper because to have sustaining value, the technique needs to be widely applicable.

2. A more quantitative assessment of sensitivity of this technique must be made. This can be done by using literature data for the desorption energies for CO and O₂ on Pt, which is well studied on single crystals. The coverage of CO and the O₂- or O₂₂- species can be calculated for a given steady state pressure and temperature based on single crystal work. Studies of pure CO and pure O₂ as a function of T and P would establish these sensitivity limits. Indeed, it is surprising that neither O₂- or O₂₂- were observed on the pure Pt. For this reason, studies of pure O₂ on the Pt particles is important to establish the credibility of the measurements.

3. The authors assert that CO blocks the adsorption of the dioxygen species. Quantitative studies as a function of CO:O₂ ratio would establish whether the argument about CO poisoning is correct.

4. The authors did not mention possible contributions of Fe-O vibrations in the Raman spectrum that would indicate the presence of atomic O on the catalyst. It is likely that the Fe is oxidized by exposure to O₂ so that atomic O co-exist with the O₂- and O₂₂- that are observed spectroscopically.

5. The authors use activity for CO oxidation as a probe for possible interaction of the Pt and PtFe nanoparticles with the underlying plasmonic Au particle. This does not really demonstrate that there is no effect of the Au/silica “shiner” because CO oxidation occurs readily on Au if O is present. XPS of the Au 4f region should be reported to look for changes due to coating the “shiner” with the catalyst, both before and after catalytic reaction.

6. It is unclear what the state of the PtFe nanoparticle is under catalytic conditions. The *ex situ* XPS cannot be used to identify catalytically relevant species present under reaction conditions. It is critical to point out the difference in *ex situ* vs. *in situ* XPS. Therefore, some of the claims made should be modified. While there is some evidence for Fe oxidation in the *ex situ* XPS, the authors refer to the

particles as alloys. The state of the material most probably depends on the O₂:CO ratio during reaction. At sufficiently high oxygen chemical potential, the Fe will be oxidized and it is likely to phase separate from the Pt. This point warrants discussion and the XPS data shown in Fig. 3a needs to be evaluated in more detail via curve fits and quantitative analysis. The peaks for the alloy particles are very broad and illdefined and the interpretation of the data is simplistic. Besides changes in oxidation state, chemical shifts occur due to alloying and also due to final state effects in small particles (See review by HJ Freund on metal nanoparticles on alumina thin films for details on final state effects.) The alloying effect could be assessed by comparison to the literature or by studying an alloy thin film. Likewise, the authors claim that *ex situ* XPS of the Pd particles shows "...existence of surface oxygen species on Pd nanocatalysts..." First of all the data shown in Fig. S12 is not definitive because there is only a subtle change in peak shape. Second, the material will not be the same *ex situ* as under reaction conditions. Overall, the understanding of the materials properties under reaction conditions is lacking.

7. The catalytic performance for CO oxidation on the Pd nanoparticles is more limited and, therefore, the interpretation cannot be judged. What is the composition of the gas mixture for CO oxidation on the Pd system in Fig. 4? How does this compare to known literature data? As described above for Pt, the steady state coverages for the CO and the oxygen species can be estimated using single crystal data. Rather than invoke Eley-Rideal mechanisms for Pd, it is possible that the steady state coverage of CO under reactions is too low to be detected. It could be that CO only binds to defects in the PdO layer and that they are very reactive. Generally, the Pd work is not as strong as the Pt and PtFe. The authors should consider a more complete exposition of the Pd work elsewhere and focus instead here on the Pt cases. Otherwise, the Pd work needs to be described better and in more detail, not just using DFT.

8. The DFT is not benchmarked to the experiments so there is no way of evaluating its validity. It is not clear that it adds value.

More minor, but important details:

9. First paragraph of the Results section states that "...the enhanced electromagnetic field generated by the shell-isolated nanoparticles...decreases exponentially." This needs to be explain more rather than just asserted. Similarly, in the early part of the Results section, simulations using 3D-FDTD methods are referred to as the basis of Fig. 1. More specifics regarding these simulations and the assumptions made in them should be included. A brief description and set of assumptions should be in the main text; more detailed equations should be provided in supplementary material. This is important to establish the validity of the technique. (The short description in SI is not really adequate.)

10. The authors state that the silica films are not porous? There are different length scales of porosity in silica. How was the porosity (or lack thereof) established?

11. It is surprising that the dioxygen species would persist on the surface *ex situ*, as is claimed based on the EPR. A more detailed interpretation of the EPR is required to support this point. There is clearly an EPR signal, but it is not obvious that it is due to "active oxygen" as claimed. What changes might be expected if, for example, oxidized magnetite (Fe₃O₄) particles were formed under oxidizing conditions? Could this explain the EPR?

12. How were the catalysts pretreated?

13. References to those who have made progress in understanding catalysis is too narrow. Only a few papers are cited and from a limited subset of groups. The major contributions by others, e.g. Madix and Freund, should be included. I also suggest citing review articles rather than a few narrow papers. Finally, the value of other *in situ* methods should be mentioned. There is a recent Chem. Rev. article on this topic by Crozier and Tao that would be a good citation to include.

14. There are no page numbers on the pdf file making it very difficult to refer to specific parts of the paper. In revision, authors should be sure there are page numbers.

Response to Reviewer 1.

Reviewer #1 (Remarks to the Author):

This manuscript describes measurement of surface enhanced Raman spectra (SERS) during CO oxidation catalysis on metal catalysts (e.g., Pt, FePt, Pd) supported on ~55nm Au nanoparticles coated with a 2nm thick SiO₂ layer. The Au nanoparticles create surface plasmon resonance (SPR), which enhances the Raman signal, while the SiO₂ layer acts as an inert catalyst support and isolates the Au from reactants and catalysts. This SiO₂ shell isolation technique allows extending the SERS to general systems beyond metals like Au that exhibit SPR, which has been described in detail in previous publications from the authors. The novelty of the present work is to demonstrate that the vibrations from metal-C bonds and various surface oxygen species (oxo, peroxy, superoxy) during CO oxidation can be probed using this technique, which allows to confirm certain differences in reaction mechanisms on different catalysts. The prepared materials are characterized using microscopy and various spectroscopic methods, and some conclusions are supported by DFT calculations. This work is of general interest in catalysis for the detection of low frequency vibrational modes during reactions, and is publishable if the issues mentioned in the comments below can be addressed.

Response: We thank the reviewer very much for the positive comments.

(1) The presence of surface oxygen species detected by the authors is qualitatively consistent with the proposed mechanism. However, these measurements were performed at one fixed reactant concentration in the feed, and only one temperature for Pt and FePt. A broader range of reaction conditions must be used to demonstrate, which of the species concentrations change linearly with reaction rates, or in a manner expected from proposed mechanism.

Response: Thanks for the reviewer's suggestion. To better understand the reaction process and mechanism for CO oxidation over Pd nanocatalysts, the reactions have been studied under different CO/O₂ ratios by our in-situ SHINERS-satellite strategy in the revised manuscript. As shown in Figure R1, only CO is adsorbed on the catalysts at low temperatures for all CO/O₂ ratio. The Raman signals for oxygen species and PdO_x arise while those for adsorbed CO species decline with the increase of reaction temperature. These results indicate that the surface reaction process of CO oxidation on Pd nanocatalysts under different feed conditions is similar. However, the temperature when surface oxygen species and PdO_x appear as well as the ratio of the Raman intensity of adsorbed CO to their intensity increases with the CO/O₂ ratio of the feed. At the same time, the activity of CO oxidation decreases with the increase of the CO/O₂ ratio (Figure R1d). These results give further evidence implying that surface oxygen species and PdO_x are essential for the CO oxidation on Pd nanocatalysts, and CO will inhibit the activation of O₂ on the catalysts

surface. Therefore, the oxygen species form at higher temperatures and the activity decreases, as CO/O₂ ratio increases. It should also be noted that almost no CO is observed on the catalysts at high temperatures where catalytic efficiency is high for the CO/O₂ ratio of 1/10, and 1/1, while both CO and O₂ are adsorbed on the catalysts under similar conditions for the CO/O₂ of 5/1. This means CO oxidation on Pd nanocatalysts at high temperature will change from E-R to L-H mechanism with the increase of CO/O₂. This can be explained by the DFT calculations that the adsorption energies of CO and O₂ on PdO_x are comparable (Table S1 and S2), so that the surface coverage of CO and O₂ are dependent on their partial pressure. According to the above findings, we have revised Figure 4 as well as the relevant discussion in our revised manuscript.

Figure R1. SHINERS-satellite spectra for CO oxidation over Pd nanocatalysts under different feed conditions. (a) CO/O₂=1/10, (b) CO/O₂=1/1, (c) CO/O₂=5/1. (d) Catalytic performance of CO oxidation over Pd nanocatalysts under different feed conditions.

As for the Pt-based system, we also studied the adsorption of CO and O₂ on Pt nanocatalysts as a function of temperature using in-situ SHINERS-satellite spectroscopy. As shown in Figure R2a, there are only Raman bands for the Pt-C stretch when CO is adsorbed on the catalysts. No new Raman band appears, and the intensity of the Raman bands for the Pt-C stretch decreases rapidly, as temperature increases. This means CO desorbs from the catalyst surface with the

increase of temperature, leading to lower CO coverage at higher temperatures. However, Raman bands for oxygen species appear when the feed is changed to pure O₂ after CO adsorption (Figure R2b). This result indicates that oxygen can be easily adsorbed on the catalyst surface if only O₂ is present in the feed. With the increase of reaction time and temperature, the intensities of oxygen species increase as those for Pt-C stretch decrease. Compared with the in-situ SHINERS-satellite spectra during CO oxidation at 30 °C in Figure 3 of main text and Figure R2c which only show the Raman bands for Pt-C stretch, it can be concluded that the activation of O₂ would be inhibited by the gas phase CO in this case. This phenomenon can be explained by the DFT calculations of Bao et al.¹ They found the adsorption energy for CO on Pt (-1.64 eV) is much more negative than that for O₂ on Pt (-0.71 eV). Thus, the surface of Pt tends to be covered by CO, resulting in blocking the sites for O₂ adsorption. The inhibition of O₂ adsorption by CO can be further confirmed by the SHINERS-satellite spectrum of CO oxidation at 100 °C (Figure R2c). As illustrated by the temperature dependent SHINERS-satellite spectra of CO adsorption (Figure R2a), CO will desorb from the Pt surface at this temperature, thus releasing active sites for the O₂ adsorption. Therefore, Raman signals for CO and O₂ can be observed at the same time during CO oxidation at high temperatures, so as to enhance the activity. On the other hand, the adsorption energies for CO and O₂ are comparable (1.30 eV for CO and 1.51 eV for O₂, respectively) for PtFe bimetallic catalysts¹. Therefore, CO and O₂ can be co-adsorbed on the surface simultaneously even at room temperature, leading to the high activity of PtFe bimetallic catalysts (Figure 3b). According to these complementary experiments data, we have revised the discussion about Figure 3 in the main text, and added Figure R2 to the Supplementary Information as Supplementary Fig. 13.

Figure R2. In situ SHINERS-satellite spectra of CO or O₂ adsorption on Pt nanocatalysts as a function of temperature. (a) CO adsorption on Pt nanocatalysts. (b) CO was first adsorbed on the catalysts, then the feed was changed to pure O₂. (c) in-situ SHINERS-satellite spectra of CO oxidation on Pt nanocatalysts.

(2) It seems likely to this reviewer, that the material synthesis methods leave some strongly

bound chemical species on the catalyst surfaces. It is not clear if some treatment was used to remove those species and a complete access of catalyst sites to reactants was confirmed using chemisorption or any other methods.

Response: The catalysts are synthesized using oleylamine or oleic acid as capping agents in order to obtain desired shape, size and composition. These capping agents are then exchanged by NOBF_4 in the procedure of surface modification. NOBF_4 binds to the surface much more weakly, and can be easily replaced or removed². Furthermore, the catalysts-on-SHINs nanocomposites are treated under H_2 before the in situ SHINERS study, which will further avoid the catalyst surface contamination. This can be checked by the XPS study of the catalysts. As shown in Figure R3, a clear XPS signal for N 1s is observed on the as-prepared Pt nanocatalysts, which is from the NH_2 -group of the oleylamine adsorbed on the catalyst. This signal disappears on the treated catalyst, indicating the oleylamine molecules are removed from the catalyst surface after the treatments. Therefore, it can be concluded that the catalyst sites are accessible to reactants. Figure R3 has been added to the SI as Supplementary Fig. 11 in the revised manuscript, and the discussion in page 7 of the main text has also been revised accordingly.

Figure R3. XPS spectra of Pt catalysts before and after treatments.

(3) In the last paragraph of page 8 the reviewers mention that 360 cm^{-1} peak on Pd is from Pt-C vibrations of bridge CO, but a corresponding peak from linear CO with the same intensity ratio as that for the C-O stretch for the two species seems missing from the spectra. Is there a reason why this peak is missing?

Response: We greatly appreciate the reviewer's comments. It is well known that CO mainly adsorbed on Pd surface via bridged configurations, as the adsorption energies for them are much larger than those for linear configurations³. Therefore, the intensity of the Raman bands for

bridged C-O stretch at 1935 cm^{-1} are much stronger than those for linear C-O stretch at 2061 cm^{-1} . As for the low wavenumber region, there is a very small peak at around $480\text{-}490\text{ cm}^{-1}$, besides the peak for the Pd-C stretch of bridged CO, which can be assigned to the Pd-C stretch of linearly adsorbed CO. As the reviewer comments, the ratio between the linear Pd-C stretch and the bridge Pd-C stretch is much smaller as compared with that for C-O stretches. Similar results have also been reported for adsorption on Pd coated Au films by Weaver et al.⁴ This could result from the fact that the Raman scattering cross section for linear Pd-C stretch is smaller than that for bridged Pd-C stretch. The discussion in page 9 has been revised as noted above.

(4) The portion of spectra and conditions shown for the Pt and FePt samples are different from that shown for the Pd samples. The Supporting Information on comparing them at same spectral features and details of reaction dynamics should be provided to confirm internal consistency.

Response: According to the reviewer's suggestion, Supplementary Figure 21 has been added in the SI to compare the SHINERS-satellite spectra of CO oxidation on Pt, PtFe, and Pd nanocatalysts. As shown in Supplementary Figure 21, CO is mainly adsorbed on Pt via linear configuration, while on Pd via bridge configuration. This is because the adsorption energy for linear CO on Pt is larger than that for bridge CO, while that for Pd catalysts just behaves oppositely³. The spectra also show that only Raman bands for Pt-C or Pd-C stretch modes can be observed for CO oxidation on Pt and Pd at $30\text{ }^{\circ}\text{C}$, but no signals for oxygen species can be observed. This is because CO adsorption on Pt and Pd is so strong that almost all the catalyst surfaces are covered by CO at this temperature, leaving nearly no active sites for O_2 activation. Thus, the active sites for O_2 activation are blocked. As for PtFe bimetallic catalyst, the adsorption energies for CO and O_2 are comparable due to the promotional effect of Fe species¹. Then, CO and O_2 can be co-adsorbed on the catalyst simultaneously, thus the reaction between can happen even at room temperature via the L-H mechanism. As for CO oxidation on Pd nanocatalysts at high temperatures, Pd is oxidized to surface PdO_x , and CO would desorb from the catalyst surface. This will lead to the exposure of active sites, on which O_2 can then be activated. Therefore, the oxidation of CO can proceed on the catalyst via E-R mechanism at this condition. This discussion has also been added in the SI of the revised manuscript.

(5) The last paragraph on page 7 mentions that the "Raman enhancements can be preserved even after high-temperature thermal treatments (Fig. S9)." The temperature of the treatment should be mentioned in the main text, and a comparison of peaks at same conditions before and after thermal treatment should be provided in the SI.

Response: Thank you for the valuable suggestions. We have measured the Raman spectra of CO

adsorption on Pt-on-Au and Pt-on-SHINs treated under different temperatures for 30 min. All the Raman spectra are measured under the same conditions in order to reveal the effects of thermal treatments on the Raman intensity. As shown in Figure R4, strong Raman bands ascribed to Pt-C stretch of CO adsorption can be observed on the fresh Pt-on-Au. However, almost nothing appears on the Pt-on-Au treated at 300 °C. This means the Pt-on-Au shows very poor thermal stability, and its SERS activity would decrease rapidly after thermal treatment. On the other hand, the Pt-C Raman bands for CO adsorbed on Pt-on-SHINs can still be observed, though the intensities decrease with the increase of temperature. These results indicate the SHINERS-satellites nanocomposites have much better thermal stability than Pt-on-Au. According to the SEM characterizations in Figure R4, we believe this is because that coating ultrathin silica shells on Au nanoparticles can stop them from coalescing. Based on these results, we have added Figure R4 to the SI as Supplementary Fig. 12, and revised the discussion about the thermal stability in the main text and the SI.

Figure R4. Raman spectra of CO adsorption on Pt-on-Au and Pt-on-SHINs treated at different temperatures (left), and the SEM images of Pt-on-Au and Pt-on-SHINs treated at 300 °C for 30 min (right).

(6) The abbreviation SHINERS is not defined the first time it is used in the abstract.

Response: We greatly appreciate the review for the comments. We have defined the abbreviation SHINERS in the abstract of the revised manuscript.

Response to Reviewer 2

Overall, this is a very nice paper that describes the application of a relatively new and novel method for *in situ* characterization of catalytic particles. These types of studies are very important as one piece of the puzzle for understanding heterogeneous catalysis. Although the authors claim that they are using this technique to “understand” the catalysis so as to facilitate rational design, the technique by itself is limited because it only probes vibrations of species on the surface; thus, it does not directly “see” changes in the catalyst structure or composition. I strongly suggest that the authors note that their new tool is a step forward but that it needs to be combined with one or more *in situ* method (e.g. ambient pressure XPS, X-ray absorption or environmental TEM) to get a full picture of the catalyst. In other words, it is important to point out that one technique cannot solve the problem of rational design. I think this paper would be a great choice for *Nature Communications* once revised to address the general point above and the technical details below.

Response: We greatly appreciate the reviewer for the high regard of our work as well as the valuable suggestions. We strongly agree with the reviewers’ comments that catalysis, especially heterogeneous catalysis is a very complicated matter, and it can hardly be fully understood by only one technique. In this study, we are trying to establish a facile strategy to track the surface intermediates and reaction processes under working conditions by *in situ* SHINERS. Therefore, surface species whose vibrational modes locate at low wavenumber region can be easily studied, which is a great challenge of traditional method. We believe this strategy can be a great complementary method for traditional characterization techniques, such as ambient pressure XPS, X-ray absorption, environmental TEM and *in situ* IR, and help to reveal structure-activity relationships and reaction mechanisms. We strongly agree with the reviewer’s idea that this strategy should be combined with other techniques in order to gain a full/better understanding of catalysis and facilitate rational design towards desired catalysts. Actually, we have also used several other techniques including HR-TEM, XPS, EPR, etc. to characterize the structures and properties of catalysts besides *in-situ* SHINERS-satellite spectroscopy in this work. According to the reviewer’s suggestion, we have emphasized these points in the Discussion section of our revised manuscript.

(1) The authors focus entirely on CO oxidation, one of the simplest catalytic reactions. Some discussion of how generalizable this method beyond CO to other systems that may not have such strong Raman signals should be added. For example, would it be possible to study ethylene or propylene epoxidation, methanol selective oxidation, or other important reactions? What are the limits of temperature and pressure for this technique? Adding such a discussion would increase the impact of the paper because to have sustaining value, the technique needs to be widely applicable.

Response: Thanks for the reviewer’s suggestions. As the reviewer said, it would greatly increase

the impact of the SHINERS-satellite strategy reported here, if it could be widely applied to various reactions on different catalysts. As shown in Figure 2, SHINERS-satellites nanocomposites consisting of a variety of nanocatalysts including metal, nanoalloy and oxide etc. can be easily fabricated by the charge-induced self-assembly method. Therefore, Raman signals from surface species on these catalysts can be recorded by the SHINERS-satellites strategy, indicating it can be a general technique for various nanocatalysts. As a result of this suggestion, we have since studied the adsorption of other weakly adsorbed molecules such as ethylene on the catalysts. Figure R5, which has been added to the SI as Supplementary Fig. 24, shows a typical SHINERS-satellite spectrum of ethylene adsorbed on Pd nanocatalysts. The Raman bands at 1280, 1550, and 2840-3040 cm^{-1} can be attributed to the coupled C=C stretch, CH_2 scissors, and C-H stretch modes of ethylene adsorbed Pd⁴. This result indicates this strategy can also be used to track surface species and intermediates during other industrially important reactions besides the model CO oxidation. Furthermore, the Raman enhancement can be further improved by optimizing the “amplifiers” (for example, by increasing the core size or changing the Au cores to Ag cores)^{5,6}, so that the species with smaller Raman scattering cross-sections can be better detected. For example, we have also found the SHINERS-satellite nanocomposites with Ag SHINs, whose enhancement is one order of magnitude higher than that of Au SHINs⁷, could also be fabricated by the self-assembly method (Figure R6, which has been added to the SI as Supplementary Fig. 26).

Figure R5. SHINERS-satellite spectrum of ethylene adsorbed on Pd nanocatalysts.

Figure R6. TEM images of 5 nm Pt nanocatalysts-on-Ag SHINs.

At the same time, the SHINERS-satellite strategy can also be applied to characterize the surface composition and electronic properties of catalysts by studying the adsorption of probe molecules. As shown in Figure R7 (added to the SI as Supplementary Fig. 25), the compositions of the outermost atomic layers of Pd@Pt core-shell catalysts can be determined using phenyl isocyanide as probe molecule. Two Raman bands at 2020 and 2150 cm^{-1} are identified on Pd nanocatalysts, which can be attributed to the $\text{C}\equiv\text{N}$ stretch modes of bridged and linearly adsorbed phenyl isocyanide, respectively⁸. As for Pt nanocatalysts, only one Raman band at about 2170 cm^{-1} is observed, which can be assigned to the $\text{C}\equiv\text{N}$ stretch modes of linearly adsorbed phenyl isocyanide. The distinct difference of the Raman spectra of phenyl isocyanide adsorbed on Pd or Pt allow us to uncover the surface composition of PtPd bimetallic catalysts. As shown in the SHINERS-satellite spectrum on Pd@Pt core-shell catalyst (the blue curve in Figure R7), only the Raman band attributed to linearly adsorbed phenyl isocyanide species ($\sim 2170 \text{ cm}^{-1}$) can be observed, while there is no Raman band at about 2020 cm^{-1} . This means that only Pt atoms are present on the surface of Pd@Pt core-shell catalysts. Furthermore, we have also found that the Raman shift of the $\text{C}\equiv\text{N}$ stretch modes was very sensitive to the electronic properties of catalysts. It would shift to higher values with the decrease of the Pt shell thickness, indicating the stronger electronic interactions between Pt and Pd. This means the SHINERS-satellite can also be used to probe the electronic properties of catalysts if combined with XPS.

Figure R7. (a) TEM image of Pd@Pt-on-SHINs. (b) SHINERS-satellite spectra of phenyl isocyanide adsorbed on Pd, Pt and Pd@Pt core-shell nanocatalysts.

The above results and discussion demonstrates that the SHINERS-satellite strategy is not only a general method for monitoring various reactions on different catalysts, but can also be used to determine the surface structure and electronic properties if probe molecules are employed. We have added Figures R5-R7 to our supporting information as noted above, and we have added a paragraph in the Discussion section of our revised manuscript to discuss the generality and potential applications of the SHINERS-satellite strategy.

(2) A more quantitative assessment of sensitivity of this technique must be made. This can be done by using literature data for the desorption energies for CO and O₂ on Pt, which is well studied on single crystals. The coverage of CO and the O₂⁻ or O₂²⁻ species can be calculated for a given steady state pressure and temperature based on single crystal work. Studies of pure CO and pure O₂ as a function of T and P would establish these sensitivity limits. Indeed, it is surprising that neither O₂⁻ or O₂²⁻ were observed on the pure Pt. For this reason, studies of pure O₂ on the Pt particles is important to establish the credibility of the measurements.

Response: Thanks for the reviewer's comments. According to the reviewer's suggestion, we have calculated the coverage of CO and O₂ based on the competitive Langmuir adsorption model. The adsorption energies for CO and O₂ on Pt are adopted from the literature (*Science* 2010, 328, 1141)¹, and they are 1.64 eV and 0.71 eV, respectively. As shown in Figure R8a and R8b, the coverage of O₂ increases with the increase of O₂ pressure or reaction temperature. At the reaction conditions studied in our work (30 °C, P_{O₂} = 10⁴ Pa, P_{CO} = 10³ Pa), the surface of Pt is almost completely covered by CO, and the coverage of O₂ is about zero. This is because the adsorption energy for CO on Pt (-1.64 eV) is much more negative than that for O₂ on Pt (-0.71 eV), so that the surface of Pt tends to be covered by CO, which would block the sites for O₂ adsorption.

Therefore, only the Raman signals for CO was observed at these conditions (Figure 3c). According to the reviewer's suggestion, we have also studied the in situ SHINERS-satellite spectra of Pt nanocatalysts under pure CO and pure O₂ as a function of temperature. As shown in Figure R8c, there are only Raman bands for the Pt-C stretch when CO is adsorbed on the catalysts. The intensities of these Raman bands decrease rapidly but no new Raman band appears, as temperature increases. This means CO starts to desorb from the catalyst surface, leading to lower CO coverage at higher temperatures. On the other hand, Raman bands for Pt-C decline while Raman bands for oxygen species appear, if the feed is changed to pure O₂ after CO adsorption (Figure R8d). This means oxygen can be adsorbed on the catalyst surface if only O₂ is present in the feed. With prolonged reaction time and increased temperature, the intensities for the oxygen species increase while those for Pt-C stretch decrease. Compared with the in-situ SHINERS-satellite spectrum during CO oxidation on Pt in Figure 3c of main text which only shows the Raman bands for Pt-C stretch, it can be concluded that the activation of O₂ would be inhibited by the CO. This discussion has been added to the SI, together with Figure R8 (Supplementary Fig. 13), and the description on page 8 of the main text has also been revised as necessary.

Figure R8. (a), (b) The calculated coverage of O₂ and CO on Pt as a function of O₂ pressure and

temperature based on the competitive Langmuir adsorption model, respectively. The adsorption energies for CO and O₂ on Pt used in the calculation is adopted from the literature (*Science* 2010, 328, 1141)¹. (c), (d) In situ SHINERS-satellite spectra of CO or O₂ adsorption on Pt nanocatalysts as a function of temperature. (c) CO adsorption on Pt nanocatalysts. (d) CO was first adsorbed on the catalysts, then the feed was changed to O₂.

(3) The authors assert that CO blocks the adsorption of the dioxygen species. Quantitative studies as a function of CO: O₂ ratio would establish whether the argument about CO poisoning is correct.

Response: As shown in Figure R9, the adsorption energies for CO on Pd and Pt are much more negative than those for O₂. Therefore, the catalyst surface will be preferentially covered by CO, when exposed to the gas mixture of CO and O₂. This will then block the active sites for O₂ activation. This phenomenon has been well demonstrated by studies on single crystal surfaces^{9,10}. According to the reviewer's suggestion, we have also acquired the in-situ SHINERS-satellite spectra of CO oxidation on Pd or Pt nanocatalysts under different conditions. As for the Pd system (Figure 4 in the main text and Figure R10), the oxygen species appear at higher temperatures and the ratio between the Raman peak intensities for them and Pd-C decrease with the increased CO/O₂. Furthermore, the activity of CO oxidation on Pd also drops dramatically with the increase of CO/O₂ (Figure R10d). As for the Pt system, only Raman bands for adsorbed CO can be observed when CO is present in the feed gas (Figure 3c in the main text and Figure R8). However, oxygen species can be clearly observed, and their intensities will increase with prolonged reaction time and increased temperature, when only O₂ is present in the gas phase. These experimental results directly demonstrate that the adsorption and activation of O₂ would be inhibited by CO, as it will be strongly bound on the catalyst surface. These findings have been added to the main text as noted above in order to demonstrate the idea that activation of O₂ would be inhibited by the strong adsorption of CO.

Figure R9. DFT calculated adsorption energies for CO and O₂ on Pd (111) or Pt (111). The adsorption energies of CO and O₂ on Pd (111) are from the DFT calculation results in this work (Supplementary Information, Tables S1 and S2), while those on Pt (111) are adopted from the literature¹.

Figure R10. SHINERS-satellite spectra for CO oxidation over Pd nanocatalysts under different feed conditions. (a) CO/O₂=1/10, (b) CO/O₂=1/1, (c) CO/O₂=5/1. (d) Catalytic performance of CO

oxidation over Pd nanocatalysts under different feed conditions.

(4) The authors did not mention possible contributions of Fe-O vibrations in the Raman spectrum that would indicate the presence of atomic O on the catalyst. It is likely that the Fe is oxidized by exposure to O₂ so that atomic O co-exist with the O₂⁻ and O₂²⁻ that are observed spectroscopically.

Response: Actually, we did not observe any Raman bands attributed to Fe-O vibrations, which are located at about 670, 543, and 293 cm⁻¹ for Fe₃O₄¹¹, 221, 288, and 405 cm⁻¹ for Fe₂O₃¹², and 660 cm⁻¹ for FeO¹³. We believe there are two reasons for this. First, the FeO_x are amorphous oxides, as no diffraction peaks attributed to them can be observed in the XRD spectra. Therefore, these Fe-O vibrations show very small Raman scattering cross sections, and are very hard to detect¹⁴. Second, the loading of iron is very low (~0.20 wt %), which will further decrease the Raman signals for Fe-O vibrations.

(5) The authors use activity for CO oxidation as a probe for possible interaction of the Pt and PtFe nanoparticles with the underlying plasmonic Au particle. This does not really demonstrate that there is no effect of the Au/silica “shiner” because CO oxidation occurs readily on Au if O is present. XPS of the Au 4f region should be reported to look for changes due to coating the “shiner” with the catalyst, both before and after catalytic reaction.

Response: Thanks for the reviewer’s suggestions. The particle size of the plasmonic Au cores used in this work is about 55-120 nm. They are quite inert in CO oxidation, as it has been demonstrated that only Au nanoparticles with diameter less than 10 nm show high activity in this reaction^{15,16}. In addition, the silica shells are pinhole-free, and the adsorption of CO or O₂ is blocked¹⁷. In order to further illustrate the effect of SHINs on the catalysts, the electronic properties of SHINERS-satellite nanocomposites were studied by XPS. As shown in Figure R11 (which has been added to the SI as Supplementary Fig. 10), the binding energies of Au 4f or Pt 4f for the Pt-on-SHIN nanocomposites are almost the same as those for pure Au nanoparticles or Pt nanocatalysts. This means the electronic interactions between Pt and Au for Pt-on-SHINs are negligible. However, the binding energies of Au 4f shift to higher values while those for Pt 4f shift to lower values as compared with pure Au or Pt, if Pt nanocatalysts are directly assembled on the bare Au nanoparticles (Pt-on-Au). This indicates electrons transfer from Au to Pt on Pt-on-Au. From these results, it can be concluded that the silica shell component of the SHINs can isolate the electronic interactions between catalysts and Au cores, so that the intrinsic behavior of the catalysts can be preserved. This figure and the associated discussion has been added in the SI to illustrate the isolating role of the silica shell.

Figure R11. XPS spectra of Pt nanocatalysts assembled on Au nanoparticles and SHINs.

(6) It is unclear what the state of the PtFe nanoparticle is under catalytic conditions. The *ex situ* XPS cannot be used to identify catalytically relevant species present under reaction conditions. It is critical to point out the difference in *ex situ* vs. *in situ* XPS. Therefore, some of the claims made should be modified. While there is some evidence for Fe oxidation in the *ex situ* XPS, the authors refer to the particles as alloys. The state of the material most probably depends on the O₂:CO ratio during reaction. At sufficiently high oxygen chemical potential, the Fe will be oxidized and it is likely to phase separate from the Pt. This point warrants discussion and the XPS data shown in Fig. 3a needs to be evaluated in more detail via curve fits and quantitative analysis. The peaks for the alloy particles are very broad and ill-defined and the interpretation of the data is simplistic. Besides changes in oxidation state, chemical shifts occur due to alloying and also due to final state effects in small particles (See review by HJ Freund on metal nanoparticles on alumina thin films for details on final state effects.) The alloying effect could be assessed by comparison to the literature or by studying an alloy thin film. Likewise, the authors claim that *ex situ* XPS of the Pd particles shows "...existence of surface oxygen species on Pd nanocatalysts..." First of all, the data shown in Fig. S12 is not definitive because there is only a subtle change in peak shape. Second, the material will not be the same *ex situ* as under reaction conditions. Overall, the understanding of the materials properties under reaction conditions is lacking.

Response: We completely agree with the reviewer that the surface compositions, chemical states and structures of catalysts will change with the reaction conditions. Currently, a few advanced techniques have been developed to study catalysts *in situ*, such as ambient pressure XPS, X-ray absorption, environmental TEM and sum frequency generation. Among these techniques, ambient pressure XPS (or *in situ* XPS) has raised growing concerns recently. For example, Tao et al. have studied bimetallic nanocatalysts under different conditions by ambient pressure XPS, and found their surface compositions and chemical states would undergo reversible changes in response to oxidizing or reducing conditions^{18,19}. With this technique, they have also identified the active sites

for water gas shift reactions on Pt-based nanocatalysts^{20,21}. These results demonstrate that ambient pressure XPS can be a powerful method to determine the surface compositions and oxidation states of catalysts under working conditions. However, in situ study of the catalysts by ambient pressure XPS can only be realized in very limited groups and labs, as it requires synchrotron light source²². At the same time, the feed gas for CO oxidation on PtFe bimetallic catalysts in this work is in O₂-rich conditions (O₂/CO = 10/1), so that the results from ex-situ XPS can partially explain the behavior of the catalysts. Therefore, we only used ex-situ XPS to study the PtFe bimetallic catalysts in this work. We have also modified the statements in the revised manuscript to avoid misunderstandings about the results. According to the reviewer's suggestion, we have performed deconvolution of Fe 2p spectrum to analyze the chemical states more quantitatively. The curve fit results in Figure R12 (added to Fig. 3a in the main text) show both metallic and ferrous species are present in the bimetallic catalysts, and their relative proportion is about 1/10. It can also be observed that the binding energies for these species are slightly higher than those in bulk Fe foil. This may be due to the lower Fe-Fe coordination and increased Fe-Pt coordination in the bimetallic catalyst, leading to pronounced electronic interactions between Fe and Pt^{23,24}. The presence of metallic iron species can also be proved by the right shift of the XRD pattern of PtFe compared with Pt (Figure R13, added to the SI as Supplementary Fig. 7). The XRD result indicates the metallic iron species form an alloy with Pt, leading to a shrinking of the Pt lattice. Therefore, we believe the PtFe bimetallic catalyst shows an alloy structure with ferrous oxide decorated on its surface, according to the characterization results of elemental mapping, XPS and XRD. Based on these new findings, we have revised the discussion on the chemical states and structure of the PtFe bimetallic catalyst in our revised manuscript. As for the Pd-based system, the XPS spectra were acquired after exposure to pure O₂. As demonstrated by the in-situ SHINERS-satellite spectrum in Supplementary Figure 15, oxygen species can form under this condition. This can be further verified by the EPR spectrum of Pd catalyst treated under similar conditions (Figure R14, added to the SI as Supplementary Fig. 17). It shows two peaks around 2.003 and 2.014, which can be assigned to surface O₂⁻ species according to the literature^{25,26}. These data have been added to the main text and SI of the revised manuscript as noted above.

Figure R12. XPS spectra of PtFe bimetallic catalysts (we note that the red color results from an overlap of the pink (Pt 4s) and peach (Fe²⁺ 2p_{3/2}) colors).

Figure R13. XRD pattern of Pt and PtFe nanocatalysts

Figure R14. EPR spectrum from Pd catalyst after exposure to O₂.

(7) The catalytic performance for CO oxidation on the Pd nanoparticles is more limited and, therefore, the interpretation cannot be judged. What is the composition of the gas mixture for CO oxidation on the Pd system in Fig. 4? How does this compare to known literature data? As described above for Pt, the steady state coverages for the CO and the oxygen species can be estimated using single crystal data. Rather than invoke Eley-Rideal mechanisms for Pd, it is possible that the steady state coverage of CO under reactions is too low to be detected. It could be that CO only binds to defects in the PdO layer and that they are very reactive. Generally, the Pd work is not as strong as the Pt and PtFe. The authors should consider a more complete exposition of the Pd work elsewhere and focus instead here on the Pt cases. Otherwise, the Pd work needs to be described better and in more detail, not just using DFT.

Response: Thanks for the reviewer's suggestion. The feed gas has an O₂ rich composition (1% CO, 10% O₂, and N₂ balance), which is the typical condition for CO oxidation on Pd-based nanocatalysts^{27,28}. To better understand the reaction processes and mechanism for CO oxidation over Pd nanocatalysts, we have studied the reaction under different CO/O₂ ratios with our in-situ SHINERS-satellite strategy. As shown in Figure R10 (Figure 4 in the main text), only CO is adsorbed on the catalysts at low temperatures for all CO/O₂ ratios. The Raman signal intensities for oxygen species and PdO_x increase, while those for adsorbed CO species decrease, as reaction temperature increases. These results indicate the surface reaction processes of CO oxidation on Pd nanocatalysts under different feed conditions are similar. However, the temperature at which surface oxygen species and PdO_x appear as well as the ratio of the Raman intensity of adsorbed CO to their intensity increases with the CO/O₂ ratio in the feed. At the same time, the activity of

CO oxidation decreases as the CO/O₂ ratio increases (Figure R10d). These results demonstrate that surface oxygen species and PdO_x are essential for CO oxidation on Pd nanocatalysts, and CO will inhibit the activation of O₂ on the catalyst surface. Therefore, as the CO/O₂ ratio increases, the oxygen species form at higher temperatures, leading to the decreased activity. At the same time, almost no Raman signals for CO are observed on the catalysts at high temperatures where catalytic efficiency is high for the CO/O₂ ratios of 1/10 and 1/1, but both CO and O₂ are adsorbed on the catalysts under similar conditions for the CO/O₂ ratio of 1/1. This indicates the reaction mechanism for CO oxidation on Pd nanocatalysts at high temperature will change from the E-R mechanism for the O₂-rich condition to the L-H mechanism for the CO-rich condition. This can be explained by the DFT calculation results, as the adsorption energies of CO and O₂ on PdO_x are comparable (Table S1 and S2), so that the surface coverages of CO and O₂ are dependent on their partial pressures in the feed. According to the above results, we have revised Figure 4 and the discussion about it in our manuscript.

(8) The DFT is not benchmarked to the experiments so there is no way of evaluating its validity. It is not clear that it adds value.

Response: The key to improve the CO oxidation is to enhance the activation and conversion of oxygen. Computationally, it has been well documented that on the PtFe system, O₂ can be activated at the Pt/Fe interface, and then CO would react with the active oxygen species^{1,29}. However, the mechanism of CO oxidation is still obscure for the Pd system. By combining in situ Raman and DFT calculations, we showed that (i) O₂ can be adsorbed and activated at high temperature, leading to the formation of surface PdO_x; (ii) with increasing oxidation state, adsorption of O₂ becomes competitive with that of CO; (iii) the excess oxygen atoms on the Pd surface are highly reactive, and could readily react with gaseous CO to produce CO₂; (iv) the resulting oxygen vacancies prefer to be occupied by oxygen under excess oxygen conditions; (v) activation of O₂ occurred through direct O-O bond breaking rather than CO assisted O-O bond breaking. Such a comprehensive understanding would help to map out the elementary steps involved in CO oxidation over Pd.

More minor, but important details:

(9) First paragraph of the Results section states that "...the enhanced electromagnetic field generated by the shell-isolated nanoparticles...decreases exponentially." This needs to be explained more rather than just asserted. Similarly, in the early part of the Results section, simulations using 3D-FDTD methods are referred to as the basis of Fig. 1. More specifics regarding these simulations and the assumptions made in them should be included. A brief description and set of assumptions should be in the main text; more detailed equations should be provided in

supplementary material. This is important to establish the validity of the technique. (The short description in SI is not really adequate.)

Response: The dependence of enhancement on the distance between the probe molecules and the Au core has been well established in our previous work (e.g., *Nature* 2010, 464, 392, *Nat. Protoc.* 2013, 8, 52)^{30,31}. As shown in Figure R15, the distance between a probe molecule (pyridine) and the Au core was tuned by changing the thickness of the silica shell. The Raman signals for pyridine adsorbed on the smooth Au substrate decrease exponentially with increasing shell thickness (distance). This experimental result agrees very well with 3D-FDTD simulations, and it is decided by the near-field natural characteristics of light-induced surface plasmons³¹. According to the reviewer's suggestion, we have revised the statement in the first paragraph of the Results section and added some references. Furthermore, we have added some important details on the FDTD method to the main text, and we have added much more detail on the FDTD method to the SI.

Figure R15. Raman spectra (a) and the corresponding intensities (b) of pyridine adsorbed on smooth Au substrate as a function of its distance to Au cores.

(10) The authors state that the silica films are not porous? There are different length scales of porosity in silica. How was the porosity (or lack thereof) established?

Response: The silica shells coated on the Au cores are prepared under optimized conditions according to our previous reports (e.g., *Nature* 2010, 464, 392, *Nat. Protoc.* 2013, 8, 52) to avoid the formation of pinholes^{30,32}. The lack of pinholes is demonstrated by the lack of adsorption on Au. As shown in Figure 4 of the main text, the Raman signals for CO adsorption on Au, which would be located at about 2130 cm⁻¹, are not observed in any spectra. This indicates that there are no pinholes in the silica shells (or if they exist, they are too small and/or too few to interfere with the experiments). Similarly, the porosity of the silica shell can also be illustrated by the adsorption of phenyl isocyanide. As shown in Figure R16, a strong Raman band at about 2205 cm⁻¹, which is attributed to the C≡N stretch modes of linearly adsorbed phenyl isocyanide on Au, can be identified when phenyl isocyanide is adsorbed on a Au surface (curve I). This band can also be observed in the SHINERS-satellite spectrum obtained using Pd@Pt core-shell catalysts if the

underlying SHINs are deliberately synthesized so that pinholes are present in the silica shell (curve II). On the contrary, only the Raman band for $\text{C}\equiv\text{N}$ stretch modes of phenyl isocyanide on Pt ($\sim 2070\text{ cm}^{-1}$) can be observed for the pinhole-free SHINs (curve III), indicating that adsorption of phenyl isocyanide on the Au core is blocked by the silica shell. Considering the dimensions of CO and phenyl isocyanide molecules, it can be concluded that if any pinholes are present in the silica shells, they are smaller than 1 nm and they would not affect the SHINERS-satellite study.

Figure R16. SHINERS-satellite spectra of phenyl isocyanide adsorption on Pd@Pt core-shell catalysts.

(11) It is surprising that the dioxygen species would persist on the surface *ex situ*, as is claimed based on the EPR. A more detailed interpretation of the EPR is required to support this point. There is clearly an EPR signal, but it is not obvious that it is due to “active oxygen” as claimed. What changes might be expected if, for example, oxidized magnetite (Fe_3O_4) particles were formed under oxidizing conditions? Could this explain the EPR?

Response: Based on the reviewer’s comments, we have since acquired the EPR spectra of PtFe catalysts under conditions similar to those for CO oxidation. Briefly, the catalysts were removed from the fixed-bed lab reactor after CO oxidation, and immediately sealed in a nuclear magnetic resonance (NMR) tube along with gas mixtures whose composition was similar to that for CO oxidation. The tube was then put in an EPR spectrometer to record the spectra of the catalysts as soon as possible, so that the surface species formed during CO oxidation could be preserved. As shown Figure R17, two narrow peaks at about 2.004, and 2.013 are observed, which can be

assigned to the O_2^- species according to the literature^{25,26}. At the same time, we do not observe the Fe^{3+} species on the catalysts, which would show a very broad peak at about $g = 2.0$. This is also consistent with the XPS results. Figure R17 has been added to the SI as Supplementary Fig. 14, and the main text has been revised accordingly.

Figure R17. EPR spectrum from PtFe catalyst under CO oxidation.

(12) How were the catalysts pretreated?

Response: The long chain molecules (e.g., oleylamine or oleic acid) on the surface of the catalysts are first replaced by the weakly adsorbed $NOBF_4$ before the fabrication of SHINERS-satellites nanocomposites. They are then treated under H_2 for 30 min in order to remove the weakly adsorbed molecules and contaminants before the in situ SHINERS study. As shown in Figure R18, a clear XPS signal for N 1s is observed on the as-prepared Pt nanocatalysts, which is from the NH_2 - group of the oleylamine adsorbed on the catalyst. This signal disappears on the treated catalyst, indicating most of oleylamine molecules are removed from the catalyst surface after the treatments. This discussion and Figure R18 (Supplementary Fig. 11) has been added to the SI of the revised manuscript.

Figure R18. XPS spectra of Pt catalysts before and after treatment.

(13) References to those who have made progress in understanding catalysis is too narrow. Only a few papers are cited and from a limited subset of groups. The major contributions by others, e.g. Madix and Freund, should be included. I also suggest citing review articles rather than a few narrow papers. Finally, the value of other *in situ* methods should be mentioned. There is a recent Chem. Rev. article on this topic by Crozier and Tao that would be a good citation to include.

Response: We greatly appreciate the reviewer's suggestions. We apologize for missing some of the important works. We have searched the literature again, and tried our best to include the most important progress on model catalysts done by surface and theoretical scientists (such as the work done by Madix et al., Freund et al., Nørskov et al., and Scheffler et. al.) in our revised manuscript. Furthermore, we have emphasized the importance of *in situ* methods in catalysis in page 1 of the revised manuscript, according to some excellent reviews including that by Tao and Crozier³³.

(14) There are no page numbers on the pdf file making it very difficult to refer to specific parts of the paper. In revision, authors should be sure there are page numbers.

Response: Thanks for the kind suggestions. We have added page numbers on our revised manuscript.

References

1. Fu, Q. *et al.* Interface-confined ferrous centers for catalytic oxidation. *Science* **328**, 1141-1144 (2010).
2. Dong, A. *et al.* A generalized ligand-exchange strategy enabling sequential surface functionalization of colloidal nanocrystals. *J. Am. Chem. Soc.* **133**, 998-1006 (2011).
3. Abild-Pedersen, F. & Andersson, M. P. CO adsorption energies on metals with correction for high coordination adsorption sites — a density functional study. *Surf. Sci.* **601**, 1747-1753 (2007).
4. Park, S., Yang, P., Corredor, P. & Weaver, M. J. Transition metal-coated nanoparticle films: vibrational characterization with surface-enhanced Raman scattering. *J. Am. Chem. Soc.* **124**, 2428-2429 (2002).
5. Chen, S. *et al.* Electromagnetic enhancement in shell-isolated nanoparticle-enhanced Raman scattering from gold flat surfaces. *J. Phys. Chem. C* **119**, 5246-5251 (2015).
6. Ding, S. Y. *et al.* Nanostructure-based plasmon-enhanced Raman spectroscopy for surface analysis of materials. *Nat. Rev. Mater.* **1**, 16021 (2016).
7. Li, C. Y. *et al.* “Smart” Ag nanostructures for plasmon-enhanced spectroscopies. *J. Am. Chem. Soc.* **137**, 13784-13787 (2015).
8. Zhong, J. H. *et al.* Probing the electronic and catalytic properties of a bimetallic surface with 3 nm resolution. *Nat. Nanotechnol.* **12**, 132-136 (2017).
9. Toyoshima, R. *et al.* Active surface oxygen for catalytic CO oxidation on Pd(100) proceeding under near ambient pressure conditions. *J. Phys. Chem. Lett.* **3**, 3182-3187 (2012).
10. Freund, H. J., Meijer, G., Scheffler, M., Schlogl, R. & Wolf, M. CO oxidation as a prototypical reaction for heterogeneous processes. *Angew. Chem. Int. Ed.* **50**, 10064-10094 (2011).
11. Dong, X., Li, L., Zhao, C., Liu, H. K. & Guo, Z. Controllable synthesis of RGO/Fe_xO_y nanocomposites as high-performance anode materials for lithium ion batteries. *J. Mater. Chem. A* **2**, 9844-9850 (2014).
12. Wang, L. *et al.* Electrospun hollow cage-like α -Fe₂O₃ microspheres: synthesis, formation mechanism, and morphology-preserved conversion to Fe nanostructures. *CrystEngComm* **16**, 10618-10623 (2014).
13. de Faria, D. L. A., Silva, S. V. & de Oliveira, M. T. Raman microspectroscopy of some iron oxides and oxyhydroxides. *J. Raman Spectrosc.* **28**, 873-878 (1997).
14. El Mendili, Y., Bardeau, J. F., Randrianantoandro, N., Grasset, F. & Grenèche, J. M. Insights into the mechanism related to the phase transition from γ -Fe₂O₃ to α -Fe₂O₃ nanoparticles induced by thermal treatment and laser irradiation. *J. Phys. Chem. C* **116**, 23785-23792 (2012).

15. Haruta, M. Size- and support-dependency in the catalysis of gold. *Catal. Today* **36**, 153-166 (1997).
16. Haruta, M., Kobayashi, T., Sano, H. & Yamada, N. Novel gold catalysts for the oxidation of carbon monoxide at a temperature far below 0 °C. *Chem. Lett.* **16**, 405-408 (1987).
17. Anema, J. R., Li, J. F., Yang, Z. L., Ren, B. & Tian, Z. Q. Shell-isolated nanoparticle-enhanced Raman spectroscopy: expanding the versatility of surface-enhanced raman scattering. *Annu. Rev. Anal. Chem.* **4**, 129-150 (2011).
18. Tao, F. *et al.* Reaction-driven restructuring of Rh-Pd and Pt-Pd core-shell nanoparticles. *Science* **322**, 932-934 (2008).
19. Tao, F. *et al.* Evolution of structure and chemistry of bimetallic nanoparticle catalysts under reaction conditions. *J. Am. Chem. Soc.* **132**, 8697-8703 (2010).
20. Zhang, S. *et al.* WGS catalysis and in situ studies of CoO_{1-x}, PtCo_n/Co₃O₄, and Pt_mCo_m/CoO_{1-x} nanorod catalysts. *J. Am. Chem. Soc.* **135**, 8283-8293 (2013).
21. Zugic, B., Zhang, S., Bell, D. C., Tao, F. & Flytzani-Stephanopoulos, M. Probing the low-temperature water-gas shift activity of alkali-promoted platinum catalysts stabilized on carbon supports. *J. Am. Chem. Soc.* **136**, 3238-3245 (2014).
22. Tang, Y., Nguyen, L., Li, Y., Wang, N. & Tao, F. Surface of a catalyst in a gas phase. *Curr. Opin. Chem. Eng.* **12**, 52-61 (2016).
23. Xu, H., Fu, Q., Yao, Y. & Bao, X. Highly active Pt-Fe bicomponent catalysts for CO oxidation in the presence and absence of H₂. *Energy Environ. Sci.* **5**, 6313-6320 (2012).
24. Ma, T. *et al.* Reversible structural modulation of Fe-Pt bimetallic surfaces and its effect on reactivity. *ChemPhysChem* **10**, 1013-1016 (2009).
25. Fernández-García, M. *et al.* New Pd/Ce_xZr_{1-x}O₂/Al₂O₃ three-way catalysts prepared by microemulsion: part I. Characterization and catalytic behavior for CO oxidation. *Appl. Catal., B* **31**, 39-50 (2001).
26. Priebe, J. B. *et al.* Water reduction with visible light: synergy between optical transitions and electron transfer in Au-TiO₂ catalysts visualized by in situ EPR spectroscopy. *Angew. Chem. Int. Ed.* **52**, 11420-11424 (2013).
27. Chen, M. S. *et al.* Highly active surfaces for CO oxidation on Rh, Pd, and Pt. *Surf. Sci.* **601**, 5326-5331 (2007).
28. Russell, A. & Epling, W. S. Diesel oxidation catalysts. *Catal. Rev.* **53**, 337-423 (2011).
29. Chen, G. *et al.* Interfacial effects in iron-nickel hydroxide-platinum nanoparticles enhance catalytic oxidation. *Science* **344**, 495-499 (2014).
30. Li, J. F. *et al.* Shell-isolated nanoparticle-enhanced Raman spectroscopy. *Nature* **464**, 392-395 (2010).
31. Taflove, A. & Hagness, S. C. *Computational Electrodynamics: the Finite-Difference*

Time-Domain Method. (Artech House Press, 2005).

32. Li, J. F. *et al.* Surface analysis using shell-isolated nanoparticle-enhanced Raman spectroscopy. *Nat. Protoc.* **8**, 52-65 (2013).
33. Tao, F. & Crozier, P. A. Atomic-scale observations of catalyst structures under reaction conditions and during catalysis. *Chem. Rev.* **116**, 3487-3539 (2016).

REVIEWERS' COMMENTS:

Reviewer #1 (Remarks to the Author):

The authors have adequately addressed the comments I provided in the previous version. I now recommend the manuscript for publication.

Reviewer #2 (Remarks to the Author):

The authors have greatly improved this paper and demonstrated the value of this method for probing catalytic processes. All of the comments in the first review were thoroughly addressed. It is an excellent choice for publication in Nature Communications.

One detail that the authors should address is the DFT calculations. DFT is notoriously bad at modeling CO bonding to metals, often predicting the wrong adsorption site. (There are quite a few papers on this topic.) They should simply comment on whether their method is accurate based on literature evaluation of different functionals and methods. This can be done in supplemental material.

Reviewer #1

The authors have adequately addressed the comments I provided in the previous version. I now recommend the manuscript for publication.

Response: We thank the reviewer very much for acceptance of our manuscript.

Reviewer #2

The authors have greatly improved this paper and demonstrated the value of this method for probing catalytic processes. All of the comments in the first review were thoroughly addressed. It is an excellent choice for publication in Nature Communications.

One detail that the authors should address is the DFT calculations. DFT is notoriously bad at modeling CO bonding to metals, often predicting the wrong adsorption site. (There are quite a few papers on this topic.) They should simply comment on whether their method is accurate based on literature evaluation of different functionals and methods. This can be done in supplemental material.

Response: We greatly appreciate the reviewer for the positive comments and kind suggestion. It has been well documented that the commonly used density functionals, such as LDA, PBE, PW91 and HSE, overestimated the adsorption energies of CO on metal surface¹⁻³. RPBE functional was specially designed to remedy the notorious problem. Gajdos et al.² found that for the late transition metal surface, the predicted adsorption energies by using RPBE agreed well with the experimental values. Additionally, Schimka et al.³ pointed out that RPBE had similar accuracy to newly developed random phase approximation (RPA) for CO adsorption on top site of Pt(111) and Rh(111). Our test calculations showed that CO is preferentially adsorbed on the hollow site of Pd(111) with the adsorption energy of -1.67 eV at 1/4 monolayer (ML) coverage, in good agreement with previous theoretical prediction (-1.68 eV) and experimental value (-1.42 eV). This discussion has been added to the Supplementary Methods.

References

1. Hammer, B. *et al.* Improved adsorption energetics within density-functional theory using revised Perdew-Burke-Ernzerhof functionals. *Phys. Rev. B* **59**, 7413-7421 (1999).
2. Gajdos, M. *et al.* CO adsorption on close-packed transition and noble metal surfaces: trends from ab initio calculations. *J. Phys.: Condens. Matter* **16**, 1141–1164 (2004).
3. Schimka, L. *et al.* Accurate surface and adsorption energies from many-body

perturbation theory. *Nat. Mater.* **9**, 741-744 (2010).